# ViTacFormer: Learning Cross-Modal Representation for Visuo-Tactile Dexterous Manipulation

## Abstract

Dexterous manipulation is a cornerstone capability for robotic systems aiming to interact with the physical world in a human-like manner. Although vision-based methods have advanced rapidly, tactile sensing remains crucial for fine-grained control—particularly in unstructured or visually occluded settings. We present ViTacFormer, a representation-learning approach that couples a cross-attention encoder to fuse high-resolution vision and touch with an autoregressive tactile-prediction head that anticipates future contact signals. Building on this architecture, we devise an easy-to-challenging curriculum that steadily refines the visual-tactile latent space, boosting both accuracy and robustness. The learned cross-modal representation drives imitation learning for multi-fingered hands, enabling precise and adaptive manipulation. Across a suite of challenging real-world benchmarks, our method achieves approximately 50% higher success rates than prior state-of-the-art systems. To our knowledge, it is also the first to autonomously complete long-horizon dexterous manipulation tasks that demand highly precise control with an anthropomorphic hand—successfully executing up to 11 sequential stages and sustaining continuous operation for 2.5 minutes.

## 1 Introduction

Recent years have seen rapid advances in robotic manipulationGeng et al. (2025; 2022); Luo* et al. (2025); Geng et al. (2023a); Seo et al. (2025); Zhang et al. (2025); Geng et al. (2023b); Kuang et al. (2024; 2025); Ding et al. (2024), with behavior cloning (Shridhar et al., 2023; 2022; Team et al., 2023) emerging as a promising method for high-precision tasks in real-world settings. However, most existing work remains limited to simple hand configurationsZhao et al. (2024) and exhibits poor generalization—largely due to the underutilization of tactile sensingXu et al. (2023); Wan et al. (2023); Wang et al. (2022); Zhang et al., which is essential for fine-grained control.

While some studies have begun integrating tactile feedback into dexterous manipulation (Lee et al., 2020; Lin et al., 2025), the learned tactile representations are often shallow and underexplored. Some works focus on replicating the success of self-supervised learning to learn representations for tactile signals (Yang et al., 2024; Xu et al., 2024; Yu et al., 2023; Higuera et al., 2024; Fu et al., 2024). However, there is still a lack of an effective model that learns cross-modal representations for visuo-tactile dexterous manipulation (Qi et al., 2023; Yuan et al., 2024). We address this limitation with ViTacFormer, a unified visuo-tactile framework that enables fine-grained, generalizable manipulation through deep cross-modal representation learning.

We address this gap with **ViTacFormer**, a unified visuo-tactile framework for dexterous manipulation. Our key idea is a cross-modal representation built with cross-attention layers that fuse visual and tactile cues at every stage of the policy. Crucially, we argue, and empirically confirm, that **predicting future tactile states** is more informative than merely perceiving current ones. ViTacFormer therefore adds a dedicated tactile-prediction head that forces the shared latent space to encode actionable touch dynamics, and then auto-regressively leverage the predicted future tactile signals for generating actions.

Experiments show that learning representations from predicted tactile signals in an autoregressive manner is challenging. To address this, we propose a two-phase curriculum: during the first 75% of

training, we use ground-truth tactile inputs to stabilize the representation learning; in the final 25%, we transition to predicted tactile signals, promoting robust cross-modal reasoning.

To evaluate ViTacFormer, we construct the first comprehensive real-world benchmark for visuo-tactile dexterous manipulation, spanning both short- and long-horizon tasks. Across all benchmarks, ViTacFormer improves the success rate by roughly **50%** over strong baselines and is, to our knowledge, the first system to successfully complete very long-horizon dexterous manipulation tasks on a real robot, achieving **11 sequential stages** and sustaining continuous manipulation for over **2.5 minutes**.

In summary, our contributions include:

- A real-world experimental setup featuring bi-manual dexterous robotic hands, a teleoperation system, a high-quality dataset for real-world dexterous manipulation, and a comprehensive benchmark suite for evaluating visuo-tactile manipulation performance.

- A novel multimodal representation learning framework that integrates cross-attention for effective modality fusion, employs autoregressive modeling to forecast tactile signals, and introduces a tailored curriculum to enhance policy learning and generalization.

- We demonstrate strong results showcasing versatile and dexterous manipulation capabilities, including success in complex, long-horizon tasks. Our method outperforms strong baselines by approximately 50% in success rate and, to our knowledge, is the first to achieve very long-horizon dexterous manipulation on a real robot, completing 11 sequential stages with 2.5 minutes of continuous operation.

## 2 RELATED WORK

### 2.1 DEXTEROUS MANIPULATION

Dexterous manipulation has emerged as a critical research frontier, with applications in tasks like grasping (Wang et al., 2022; Zhang et al.; Wan et al., 2023), in-hand manipulation (Yin et al., 2023; Handa et al., 2022; Qi et al., 2022; Wu et al., 2024; Chen et al., 2024a), in-hand orientation (Chen et al., 2022), articulated object manipulation (Bao et al., 2023; Jiang et al., 2024; Chen et al., 2024b; Zhang et al., 2024; Geng et al., 2023b; 2022; 2023a), and deformable object manipulation (Li et al., 2023; Wu et al., 2020; Zhaole et al., 2024; Wang et al., 2025). Meanwhile, behavior cloning (BC) (Shridhar et al., 2023; 2022; Team et al., 2023) empowers dexterous manipulation with an end-to-end general solution.

Among BC models, diffusion policy (DP) (Chi et al., 2023) leverages diffusion models (Song et al., 2020; Ho et al., 2020) to learn the expert actions conditioned on robot observations. Since diffusion models (Song et al., 2020; Ho et al., 2020) are good at capturing the multi-modalities from diverse data inputs, diffusion policy shows promising results on robotic applications (Zhao et al., 2024). 3D diffusion policy (DP3) (Ze et al., 2024) ulitizes 3D point clouds as robot observations. It is more generalizable compared to DP since the learned representation captures geometric information from 3D data. Action chunking transformer (ACT) (Zhao et al., 2023) views BC model as a conditional variational auto-encoder. It learns the multi-modal information from diverse expert data inputs. Empirical studies (Zhao et al., 2024) show that ACT (Zhao et al., 2023) outperforms DP (Chi et al., 2023) when the collected data is limited.

Our ViTacFormer is built on top of ACT (Zhao et al., 2023), leveraging the advantage of capturing multi-modalities in diverse expert data inputs. It offers a cross-modal representation learning for visuo-tactile dexterous manipulation. The learned representation enables precise and adaptive manipulation on multifingered dexterous hands.

### 2.2 MANIPULATION WITH TACTILE SIGNALS

Prior works with tactile signals focus mainly on learning tactile representations for robotics. Some of these works focus on leveraging force values as tactile signals. For example, (Lee et al., 2020) builds some proxy tasks such as predicting robot optical flow to extract tactile representations. HATO (Lin et al., 2025) sets up a bimanual dexterous visuo-tactile manipulation system with a diffusion policy

(Chi et al., 2023) to learn the expert behaviors. Other works focus on leveraging self-supervised learning to extract rich representations specifically from high-resolution tactile images (Yang et al., 2024; Xu et al., 2024; Yu et al., 2023; Higuera et al., 2024; Fu et al., 2024). Among these works, contrastive learning (Yang et al., 2024) and masked auto-encoding (Sferrazza et al., 2024) are two popular streams to extract the representation from raw tactile images.

However, there is still a lack of an effective model that learns cross-modal representations for visuo-tactile dexterous manipulation (Qi et al., 2023; Yuan et al., 2024). Our ViTacFormer proposes a cross-attention-based auto-regressive model for future tactile forecasting and action generation. Empirical studies show that ViTacFormer unlocks the power of visuo-tactile representation for dexterous manipulation. In particular, ViTacFormer is capable of mastering long-horizon dexterous robotic tasks.

## 3 PROBLEM FORMULATION AND HARDWARE SETUP

### 3.1 PROBLEM FORMULATION

We address imitation learning for dexterous bi-manual manipulation. Given $N$ expert trajectories $\mathcal{D} = \{\tau_i\}_{i=1}^N$, where each $\tau_i = \{(o_t^i, a_t^i)\}_{t=1}^{T_i}$ consists of multimodal observations $o_t^i$, and corresponding actions $a_t^i$. Here, the multimodal observations $o_t^i$ include robot proprioception $j_t^i$, visual observations, $v_t^i$ and tactile observations $h_t^i$ from tactile fingertips. Detailed specifications of input modalities are provided in Appendix B.1.

The goal is to learn a policy $\pi_\theta$ that maps observations to actions: $a_t = \pi_\theta(o_t)$. The policy $\pi_\theta$ is trained to imitate expert behavior and is evaluated in task space, measuring success on manipulation tasks under diverse and long-horizon conditions. Detailed specifications of action outputs are provided in Appendix B.2.

### 3.2 HARDWARE SETUP

Our hardware system consists of two Realman robot arms, each equipped with a SharpaWave dexterous hand (Fig.1(a)). Each hand is anthropomorphic, featuring 5 digits with 17 degrees of freedom (DoFs). Note that it is the developing version of SharpaWave. Visual observations are captured using two wrist-mounted fisheye cameras for close-up task views and a top-mounted ZED Mini stereo camera for global scene awareness. Tactile sensing is enabled by high-resolution (320×240) tactile sensors embedded in the fingertips, developed by Sharpa.

We adopt a custom exoskeleton-based teleoperation system to collect high-quality visuo-tactile demonstrations (Fig.1(b)). The operator wears a pair of mechanical exoskeleton gloves that are mechanically coupled to the SharpaWave hands, faithfully

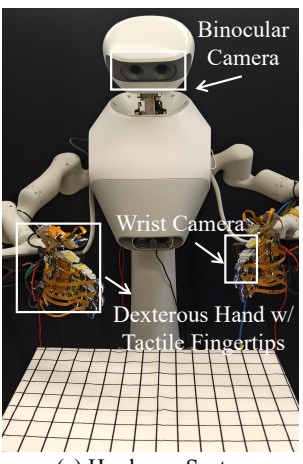
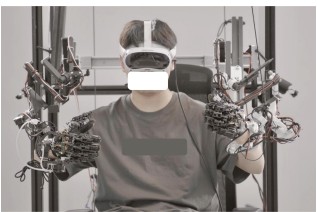

(a) Hardware System  (c) First-person VR View

(b) Human Teleoperator

Figure 1: An overview of our system hardware and teleoperation setup. (a) Our hardware system setup. (b) Teleoperator with exoskeleton gloves and VR headset. (c) VR interface with binocular and wrist views, and tactile feedback overlay.

capturing finger joint motions. A VR headset provides immersive visual feedback through a first-person interface that integrates multimodal sensory input (Fig.1(c)). The interface combines (i) a stereo top-down view from the ZED Mini, (ii) wrist-mounted local views from both arms, and (iii) real-time tactile overlays on the fingertips that highlight contact activations. This unified perception setup enables the operator to intuitively control both hands in complex, contact-rich tasks.

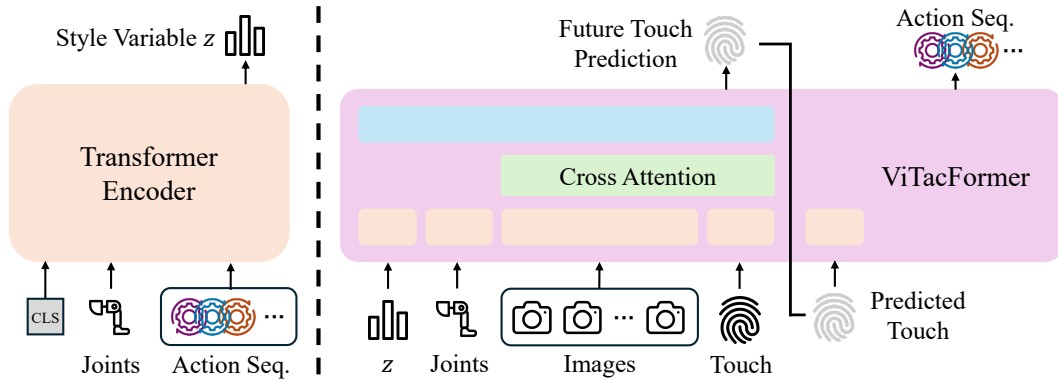

Figure 2: The neural network architecture for ViTacFormer is a conditional variational auto-encoder. Left: a transformer-based encoder maps action sequence and robot proprioception to action style variable $z$. Right: a transformer-based encoder-decoder uses style variable $z$, robot proprioception (joints), and visuo-tactile observations to auto-regressively predict future tactile signals and generate actions.

All data streams — including RGB frames, joint states, and compressed tactile maps — are time-synchronized and logged to construct multimodal expert trajectories.

# 4 METHOD

In section 4.1, we introduce a cross-attention-based multimodal integration framework that fuses the visual and tactile observation inputs. In section 4.2, we present autoregressive modeling with tactile forecasting, which better generates actions with predicted future tactile signals. In section 4.3, we summarize the network architecture and learning procedure for our ViTacFormer.

## 4.1 CROSS-ATTENTION-BASED MULTIMODAL INTEGRATION

Visual observations and tactile signals share similar semantic information. Traditional neural network architecture fuses visual and tactile observation inputs as naive token fusion. These models don't take the relevant information between visual and tactile observations into consideration.

Cross-attention is a mechanism commonly used in transformers, particularly in tasks involving multi-modal data or interacting with external knowledge. It allows the model to attend to different parts of two input sequences simultaneously, enabling it to capture interactions between them. Consequently, cross-attention-based multimodal integration motivates the agent to capture dependencies between diverse data inputs.

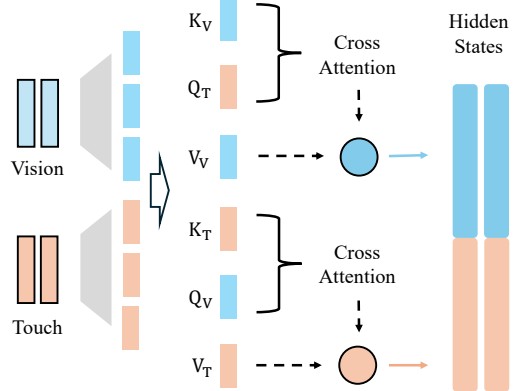

Figure 3: Cross-attention-based multimodal integration between visual and tactile observations.

In ViTacFormer, we apply cross attention to visual and tactile observations for learning better representations. This motivates the extraction of the relevant semantic information between visual and tactile signals. Fig. 3 shows the neural network architecture of multimodal integration based on cross-attention. The keys and values from visual observations are calculated with the queries from tactile signals and vice versa. Finally, the cross-attention-based features are concatenated into hidden states for further learning.

### 4.2 Auto-Regressive Modeling with Tactile Signal Forecasting

Forecasting future tactile signals motivates the agent to be aware of the change in contact signals. In detail, it motivates the latent representations involving potential future outcomes. Auto-regressively leveraging these predicted future tactile signals motivates the agent to use this prior contact knowledge for better generating actions.

To take advantages above, we formulate the action-generating procedure in two steps. First, we predict the future tactile tokens with style variables $z$, current robot proprioception (joints), and visuo-tactile observations. This tactile forecasting is stable and accurate, with an average normalized L1 error of $\approx 0.08 \pm 0.02$. Next, we concatenate the predicted future tactile signals with the current input tokens for generating actions. Note that we conduct cross-attention-based multimodal integration twice between visuo-tactile signals, both in predicting future tactile signals and generating actions.

In practice, action generation may become unstable when noisy predicted tactile signals are used as inputs in the early stage of training. To mitigate this issue, we propose a two-phase curriculum. First, we train the action generation procedure with ground-truth future tactile tokens during the initial 75% of training epochs. Finally, in the last 25% of epochs, we train the action generation procedure with predicted future tactile signals.

### 4.3 Neural Network Architecture and Learning Procedure

Fig. 2 shows the neural network architecture of our ViTacFormer. This architecture is basically a conditional variational auto-encoder. On the left of Fig. 2, there is a transformer-based encoder. It maps the robot's proprioception (joints) and expert action sequence into a style variable $z$. On the right of Fig. 2, there is a transformer-based encoder-decoder. First, it extracts the representation from visual and tactile observations with a cross-attention-based multimodal integration framework. Next, it auto-regressively predicts the future tactile signals and thus generates the actions with predicted future tactile signals. The style variable $z$ is sampled from expert demonstrations during training, while it is set to zero during inference, following ACT(Zhao et al., 2023).

We find that a combination of supervision between arm end-effectors' position and arm/hand joint angles (JA) is more effective for dexterous manipulation than arm/hand JA supervision only. The training loss of ViTacFormer is shown as:

$$\mathcal{L} = w_1 \cdot \mathcal{L}_{KL} + w_2 \cdot \mathcal{L}_{JA} + w_3 \cdot \mathcal{L}_{tactile} + w_4 \cdot \mathcal{L}_{arm}, \tag{1}$$

where $w_{1,2,3,4}$ are hyper-parmeters, $\mathcal{L}_{KL}$ is KL divergence between the action style variables and Gaussian distribution, $\mathcal{L}_{JA}$ is the $L_1$ loss based on predicted action and ground truth action, $\mathcal{L}_{tactile}$ is the $L_1$ loss based on future tactile signals and ground truth. In particular, $\mathcal{L}_{arm}$ is:

$$\mathcal{L}_{arm} = \lambda_1 \cdot \mathcal{L}_{position} + \lambda_2 \cdot \mathcal{L}_{rotation}, \tag{2}$$

where $\lambda_{1,2}$ are hyper-parameters, $\mathcal{L}_{arm}$ indicates the supervision based on arm end-effectors, $\mathcal{L}_{position}$ is the $L_2$ loss between arm end-effector's position, and $\mathcal{L}_{rotation}$ is the $L_1$ loss between arm end-effector's rotation. Empirically, we find $\mathcal{L}_{arm}$ is very useful in training dexterous manipulation skills. Detailed training hyper-parameters are provided in Appendix B.3.

## 5 Experiment

In this section, we evaluate the effectiveness of our proposed ViTacFormer. The experiments are designed to answer two questions: (1) How does our algorithm perform compared to other state-of-the-art imitation learning algorithms? The results are presented in section 5.3. (2) Is each component of our algorithm effective? The results are introduced in section 5.4.

### 5.1 Benchmark and Environment Setup

In this section, we introduce the tasks and environment setup. The tasks we conduct include 4 simple dexterous manipulation tasks and a very long-horizon visuo-tactile task. Fig. 4 shows the

| Peg Insertion | Cap Twist | Vase Wipe | Book Flip |

Figure 4: Four short-horizon visuo-tactile tasks, from left to right, i.e., peg insertion, cap twist, vase wipe, and book flip.

4 simple dexterous manipulation tasks involving peg insertion, cap twist, vase wipe, and book flip. We also conduct our ViTacFormer on a very long-horizon task, i.e., making hamburgers. Detailed task descriptions and scoring schemes are provided in Appendix C.

We conduct algorithm comparison on all tasks and ablation study on four simple tasks. These tasks range from easy to complex dexterous manipulation. The results show that our ViTacFormer outperforms other state-of-the-art imitation learning algorithms by over $50\%$ success rates. The ablation study illustrates that each component in our ViTacFormer improves the manipulation performance. Note that we use only 50 trajectories per task for training in our experiments. This makes the tasks more challenging than before.

## 5.2 METRICS AND BASELINES

**Metrics** We evaluate the algorithms with two established metrics: human normalized score and success rates. Note that the success rates may not reflect the dexterous manipulation process in detail, especially for long-horizon manipulation tasks. We define a new metric to measure the dexterous manipulation performance.

We propose Human Normalized Score (HNS) to evaluate the manipulation process in detail. First, we split the manipulation process into several stages. In each stage, we evaluate the process with $0 - 3$ raw scores. Finally, we normalize the score for a fair comparison. The HNS score is presented as:

$$\text{HNS} = \frac{\sum_{i=1}^{N} w_i \cdot s_i}{3 * \sum_{i=1}^{N} w_i},$$
(3)

which $N$ represents the number of stages in a certain manipulation task, $w_i$ represents tactile reliance in stage $i$, indicating how strongly this stage depends on tactile feedback, and $s_i$ counts from 0 to 3, which is the raw score for stage $i$ measuring its success level.

**Baselines** Diffusion Policy (DP) (Chi et al., 2023) is good at mimicking expert behaviors by capturing multi-modalities from training data with a diffusion model (Ho et al., 2020). Additionally, HATO (Lin et al., 2025) adds the tactile signals as conditions for diffusion policy (Chi et al., 2023). ACT (Zhao et al., 2023) builds a conditional variational auto-encoder for learning expert behaviors from demonstrations. ACTw/T (Zhao et al., 2023) adds the tactile signal in its input tokens. Empirical studies (Zhao et al., 2024) show that ACT (Zhao et al., 2023) outperforms DP (Chi et al., 2023) with limited training data.

In this section, we use DP (Chi et al., 2023), HATO (Lin et al., 2025), ACT (Zhao et al., 2023), ACTw/T (Zhao et al., 2023) as our baselines. Among these baselines, DP and ACT are without tactile inputs. HATO and ACTw/T take tactile signals with a naive token fusion.

## 5.3 ALGORITHM COMPARISON

*Question 1: How does ViTacFormer perform compared to other SoTA imitation learning algorithms?*

To test the effectiveness of our ViTacFormer, we conduct experiments on four short-horizon dexterous manipulation tasks. We test each algorithm on a certain task with inference 10 times. If the

results show that ViTacFormer achieves higher success rates compared to SoTA imitation learning baselines, we could prove the efficacy of our ViTacFormer.

Table 1: Success rate comparison on four short-horizon dexterous manipulation tasks. Our ViTac-Former achieves over 50% success rates compared to the baselines.

| Task | Peg Insertion | Cap Twist | Vase Wipe | Book Flip |
|---|---|---|---|---|
| DP | 2/10 | 0/10 | 3/10 | 1/10 |
| ACT | 4/10 | 4/10 | 3/10 | 2/10 |
| HATO | 4/10 | 1/10 | 4/10 | 3/10 |
| ACTw/T | 6/10 | 6/10 | 4/10 | 4/10 |
| **Ours** | **10**/10 | **10**/10 | **9**/10 | **9**/10 |

Tab. 1 shows the success rates on four short-horizon dexterous manipulation tasks. Our ViTacFormer achieves the best performance compared to other SoTA imitation learning algorithms. In particular, ViTacFormer outperforms other baselines over 50% success rates, therefore almost solving these tasks. On the other hand, tactile observation input greatly improves the manipulation performance since ACTw/T (Zhao et al., 2023) and HATO (Lin et al., 2025) outperform ACT (Zhao et al., 2023) and DP (Chi et al., 2023), respectively. More detailed inference results are provided in Appendix C.

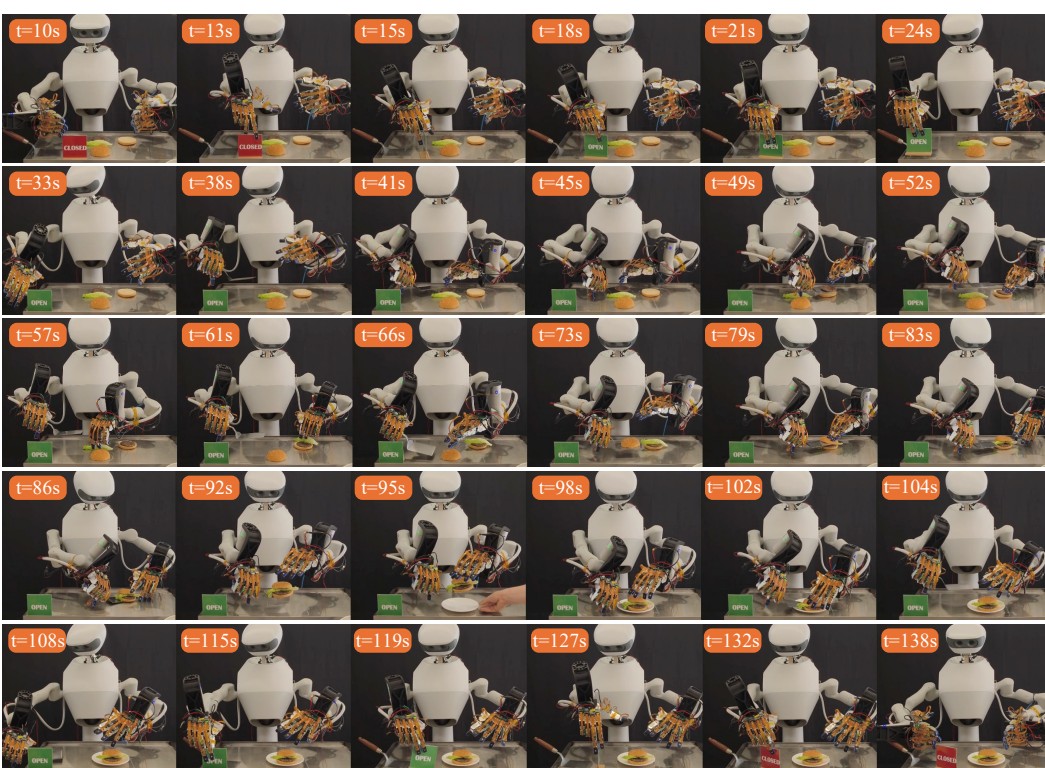

Figure 5: Successful model rollout on long-horizon task, i.e., making hamburger. We show the successful model rollout with keyframes in 11 stages. The first row represents the robot hand turning the brand to "open". The second row represents the robot hand shoveling meat to bread. The third row represents the robot hand assembling the hamburger. The fourth row represents the robot hand handing over the hamburger to the plate. The fifth row represents the robot hand turning the brand to "close".

*Question 2: How does ViTacFormer perform on complex long-horizon manipulation tasks?*

To show that our ViTacFormer is effective on long-horizon tasks, we conduct experiments on an 11-stage task, i.e., making hamburgers. To the best of our knowledge, ViTacFormer is the first system to complete very long-horizon dexterous manipulation tasks on a real robot with a single imitation

learning model. Fig. 5 shows a successful model rollout of our ViTacFormer. Our ViTacFormer masters 11 stages of making hamburgers.

Table 2: Human evaluation score comparison on a very long-horizon dexterous manipulation task. ViTacFormer shows promising results on this long-horizon task.

| Stage | 1 | 2 | 3 | 4 | 5 | 6 | 7 | 8 | 9 | 10 | 11 | Overall |
|---|---|---|---|---|---|---|---|---|---|---|---|---|
| ACT | 2.4 | 2.5 | **1.9** | **2** | 0.7 | 2.2 | 1.6 | **2.8** | 2.2 | 2.2 | 0.7 | 0.61 |
| **Ours** | **2.9** | **3** | **1.9** | 1.8 | **2.7** | **2.9** | **2** | **2.8** | **2.4** | **2.5** | **3** | **0.88** |

Tab. 2 shows the human normalized score (HNS) for each stage in the task, i.e., making hamburgers. Note that once the score is less than 1 for one stage, the model fails on this task. In experiments, we correct the mistake only when a stage fails completely (score below 1) for further stage testing. Specifically, stage 1 corresponds to the first row of Fig. 5; Stage 2-4 corresponds to the second row; Stage 5-7 corresponds to the third row; Stage 8-10 corresponds to the fourth row; and stage 11 corresponds to the fifth row. For the baseline (ACT), the model easily fails at stage 5 and stage 11, leading to almost 0 success rates on this long-horizon task. In contrast, our ViTacFormer shows promising results on all stages and achieves **over** $80\%$ **success rates** on this task.

Specifically, in stage 5 (grasping lettuce), the object is soft and deformable, so tactile sensing is required to detect contact and stabilize the grasp, while in stage 11 (flipping the sign at the end), the task demands precise timing and force control under occlusion, where tactile input enables accurate triggering of the flipping motion. These cases show that such tasks cannot be reliably solved by vision alone. More details of the long-horizon task are provided in Appendix C.2.

## 5.4 ABLATION STUDY

*Question 3: How does each component in our ViTacFormer contribute to the baseline?*

Tactile information enhances dexterous manipulation by providing force feedback, improving robustness and stability in the control process. To evaluate the contribution of each component in our ViTacFormer, we perform ablation studies by removing individual modules from the full model (Ours).

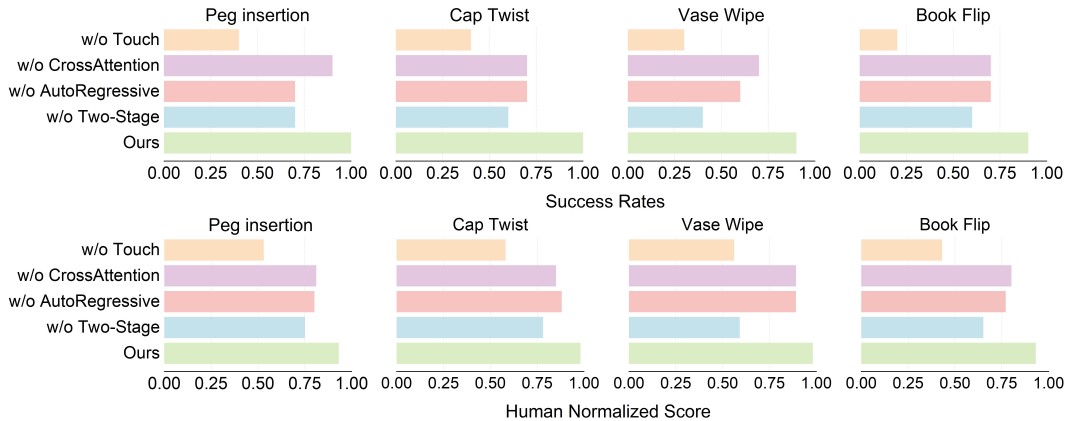

Figure 6: Ablation study. Performance comparison by removing different components from the full ViTacFormer (Ours).

Fig. 6 shows the success rates and human normalized scores for each ablated variant. Removing the cross-attention module (w/o CrossAttention) weakens the integration of visual and tactile modalities, reducing manipulation accuracy. Excluding the auto-regressive tactile forecasting (w/o AutoRegressive) impairs the temporal modeling of contact signals and reduces the information available when reasoning about actions, leading to less stable control. Dropping the two-phase curriculum (w/o Two-Stage) makes training less stable and decreases final performance. Overall, each component contributes to the performance gains, and the full ViTacFormer (Ours) achieves the best results.

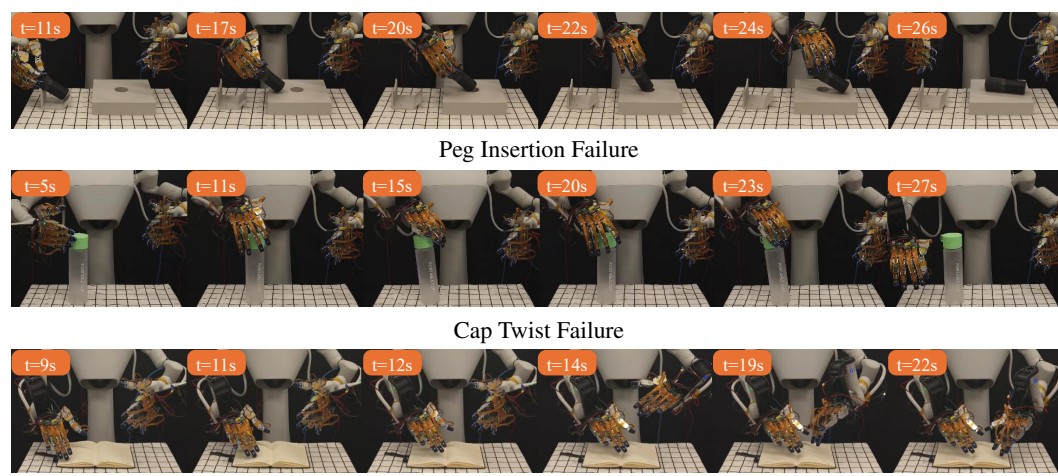

Figure 7: Failure Study. The first row is peg insertion failure, the second row is cap twist failure, and the third row is book flip failure.

*Question 4: How does the failure occur in baselines?*

There are several failure modes from the baselines. Evaluating these failure cases helps us understand the effective factors in our ViTacFormer. Fig. 7 shows the classical failure cases from the baseline ACT w/ Touch. The established tasks are highly tactile-dependent. Consequently, how to leverage the tactile signals in imitation learning is of great significance.

Fig. 7 shows the peg insertion failure. When the peg is moved to the hole, the robot hand isn't aware of the position of the hole and thus fails to insert the peg into the hole. This failure shows the importance of predicting future tactile signals in ViTacFormer. Predicting the future tactile tokens motivates the dexterous hand to be aware of the temporal force difference. When the peg is near the hole, the temporal force difference would be changed, therefore showing the hole is nearby. Consequently, it could improve the robustness of inserting the peg into the hole in ViTacFormer.

Fig. 7 shows the cap twist failure. The robot hand fails to twist the cap. The reason can be traced to the fact that the hand isn't aware of whether the cap is open or closed. It motivates the importance of the autoregressive architecture of our ViTacFormer. Auto-regressively predicting the future tactile signals and leveraging these signals simplifies reasoning about the actions under such complex situations.

Fig. 7 shows the book flip failure. The dexterous hand isn't aware of the book. It flips the book in the air. This originates from the lack of visual and tactile observation fusion. In our ViTacFormer, we use cross-attention-based multimodal integration to fuse the visual and tactile observations. This improves the performance of dexterous manipulation. More detailed failure cases are provided in Appendix C.

## 6  CONCLUSION

We present ViTacFormer, a unified visuo-tactile framework for dexterous robotic manipulation that leverages deep cross-modal representation learning. By fusing vision and touch at every stage of the policy and incorporating predictive tactile modeling, ViTacFormer enables robust, fine-grained control across a diverse set of manipulation tasks. Our curriculum-based training strategy further enhances representation stability, allowing the system to effectively reason over predicted tactile signals. Empirical results demonstrate that ViTacFormer significantly outperforms strong baselines—achieving approximately 50% higher success rates—and is the first to complete long-horizon dexterous tasks on a real robot. We believe this work opens new possibilities for generalizable, high-precision robotic manipulation through the principled integration of vision and touch.

ETHICS STATEMENT

This work investigates visuo-tactile learning for dexterous robotic manipulation. All experiments are conducted on physical robotic platforms in a laboratory environment, without involving human subjects, personal data, or sensitive information. The research is intended solely for scientific purposes, and we see no foreseeable harmful applications in its current form. We adhere to the ICLR Code of Ethics, and all authors confirm compliance with its principles throughout the research and submission process.

REPRODUCIBILITY STATEMENT

We have taken steps to ensure the reproducibility of our results. Detailed descriptions of the Vi-TacFormer architecture, training procedure, and hyperparameters are provided in the main text and appendix. Experimental setups, task descriptions, and evaluation protocols are described to enable independent replication. The full source code and instructions for reproducing all experiments are included in the supplementary file.

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

APPENDIX OVERVIEW

**Appendix A.** Use of large language models (LLMs) in this work.

**Appendix B.** Implementation and training details, including sensor modalities, action formats, and inference settings.

**Appendix C.** Expanded experimental results on all tasks, including task description, scoring schemes, inference results and failure analysis.

**Appendix D.** Limitations of our work and directions for future research.

## A    USE OF LARGE LANGUAGE MODELS (LLMS)

In this work, we used large language models (LLMs) solely as assistive tools for polishing the writing and improving the readability of the manuscript. LLMs were not involved in research ideation, experimental design, data collection, data analysis, or any other scientific contributions. All technical content, methodology, experiments, and conclusions are the original work of the authors. The authors take full responsibility for the contents of this paper.

## B    ADDITIONAL METHOD DETAILS

### B.1    INPUT MODALITIES

Our model takes multimodal inputs from the robot system, including visual observations, robot proprioception, and tactile signals.

**Visual Input**

We use four synchronized camera views as visual input: a stereo pair (180×320) from top-mounted ZED Mini cameras (Fig. 8(**a**), (**c**)), and two fisheye wrist-mounted views (256×280) for left and right hands (Fig. 8(**b**), (**d**)). All frames are encoded into image tokens via a vision backbone before cross-modal integration.

**Proprioception Input**

The robot's internal state at each timestep is represented by a 58-dimensional vector, consisting of: 7-DoF left arm state, 17-DoF left hand state, 7-DoF right arm state, 17-DoF right hand state, and 2-DoF neck state—structured as [7, 17, 7, 17, 2]. A temporal horizon of 6 frames is used, resulting in a proprioceptive input of shape (6, 50).

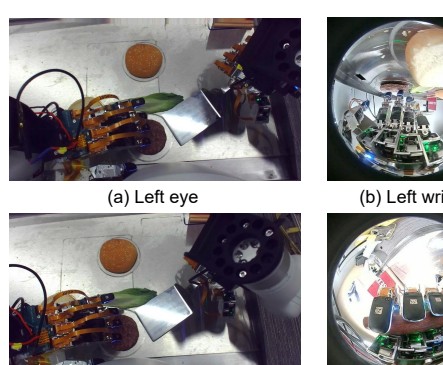

(a) Left eye          (b) Left wrist

(c) Right eye         (d) Right wrist

Figure 8: **Four types of camera views**

**Tactile Input**

Each of the 10 fingertips is equipped with force and torque sensors along 3 axes, resulting in 20 tactile channels. For each channel, we collect 18 frames of data ([18, 3]), which are concatenated into a raw tactile tensor of shape [18, 60]. We additionally compute frame-wise deltas to obtain relative changes ([18, 60]), and concatenate them with the raw signal to produce the final tactile input of shape [18, 120].

### B.2    ACTION OUTPUT

The policy generates high-frequency action sequences with shape (100, 50) per rollout, where 50 corresponds to the full control dimension of the robot: 7-DoF left arm, 17-DoF left hand, 7-DoF right arm,17-DoF right hand, and 2-DoF neck—matching the structure of the proprioceptive state. The 100-frame horizon supports fine-grained dexterous motion across extended manipulation stages.

## B.3 Data and training details

We train each task using 50 expert demonstrations and 100 epochs on 2 NVIDIA H20 GPUs. Short-horizon tasks typically converge within half a day, while long-horizon tasks (e.g., Make Hamburger) require up to 2 days. The model is optimized using the Adam optimizer with a learning rate of 1e-4 and a batch size of 128. Training supervision includes KL divergence on latent action style, L1 losses on both predicted actions and tactile signals, and auxiliary supervision on end-effector positions and rotations. All input modalities are temporally aligned and normalized prior to training.

## B.4 Inference details

During deployment, the policy runs at 10Hz, producing a 100-frame $(100, 50)$ high-frequency action sequence at each decision step. To ensure smooth and physically stable execution, we apply temporal smoothing over the predicted action trajectory before sending commands to the robot. The system is deployed on a real dual-arm platform with synchronized visuo-tactile observation streams and low-latency control.

## C Additional Experiment Details

### C.1 Short-horizon tasks

The four short-horizon tasks share a standardized tabletop workspace and a common set of objects, as shown in Fig. 9(**a**). The workspace is discretized using a printed grid (5cm per square), with the top-left corner defined as the origin $(0, 0)$, as illustrated in Fig. 9(**b**). During training, each object is placed at a designated grid coordinate. For generalization, we randomly perturb the object's position within a circular region of half-grid radius (i.e., 2.5cm) around its original anchor point.

(a) Objects  (b) Workspace

Figure 9: **Short-horizon task setup.** (a) All four short-horizon tasks share a common set of objects. (b) The tabletop workspace marked with a grid.

#### C.1.1 Peg Insertion

**Task Description**

The robot uses its right hand to grasp a cylindrical peg from the vertical rack, then moves it diagonally along the sloped platform toward the insertion hole. Upon reaching the vicinity of the hole, the robot is expected to insert the peg smoothly and stably into the hole. This task involves visual alignment, precise grasping, and tactile-guided insertion. Representative execution frames are shown in the first row of Fig. 10.

**Scoring Scheme**

Table 3: **Scoring criteria for Peg Insertion.**

| Stage 1: Grasp (weight 1) | Description |
| --- | --- |
| 0 | No grasp |
| 1 | Grasped but slipped or dropped |
| 2 | Poor or tilted grasp |
| 3 | Stable grasp |

| Stage 2: Insertion (weight 2) | Description |
| --- | --- |
| 0 | No insertion |
| 1 | Misaligned, dropped |
| 2 | Partial insertion |
| 3 | Fully inserted |

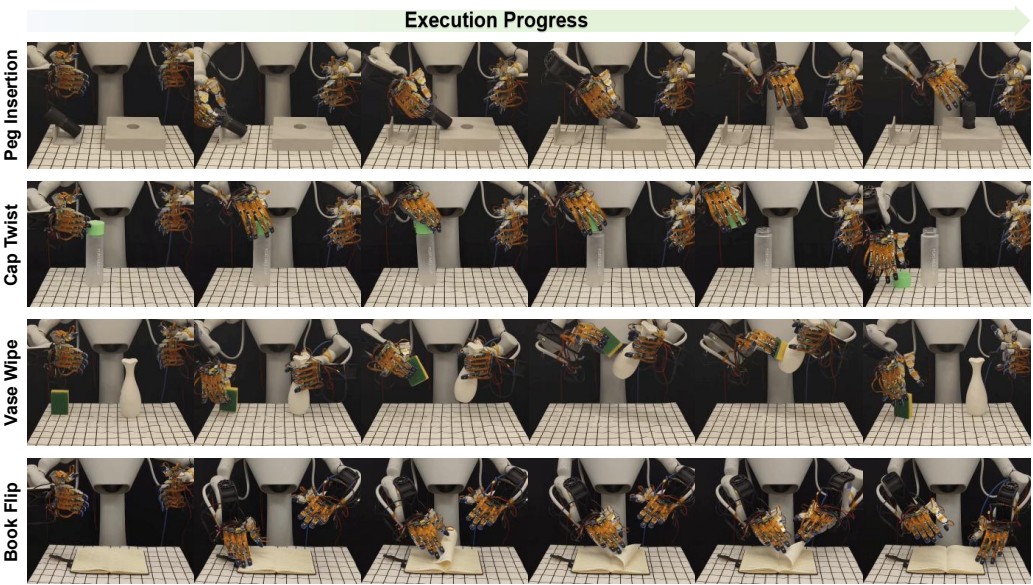

Figure 10: **Execution examples for short-horizon tasks.** Representative keyframes from four tasks: peg insertion, cap twist, vase wipe, and book flip. Each task demonstrates a full execution sequence from perception to manipulation.

The task is divided into two stages: peg grasping (weight 1) and insertion (weight 2). Each stage is scored from 0 to 3 based on qualitative criteria such as grasp stability and insertion completeness. The human normalized score (HNS) is computed as a weighted average. A total score of 3 for stage 1 and $\geq 2$ for stage 2 is considered successful.

**Inference Results**

Table 4: **Peg Insertion: inference results across models.**

| Model | Stage 1 | Stage 2 | HNS | Success Rate |
|---|---|---|---|---|
| DP | 1.6 | 0.9 | 0.37 | 20% |
| ACT | 2.6 | 1.1 | 0.53 | 40% |
| HATO | 2.4 | 1.1 | 0.51 | 40% |
| ACT w/T | 2.6 | 1.8 | 0.68 | 60% |
| Ours w/o CrossAttention | 2.9 | 2.2 | 0.81 | 90% |
| Ours w/o AutoRegressive | 3.0 | 2.1 | 0.80 | 70% |
| Ours w/o Two-Stage | 2.8 | 2.0 | 0.75 | 70% |
| Ours | 3.0 | 2.7 | 0.93 | 100% |

Table 4 summarizes the quantitative performance on the peg insertion task. We report the average stage-wise scores, human normalized score (HNS), and success rate across baselines and ablations. Our method achieves the highest HNS (0.93) and 100% success rate, demonstrating strong performance across both stages.

**Failure Case Analysis**

Figure 11, first row, shows two representative failure cases in the peg insertion task. In the first case, the robot fails to locate the insertion hole accurately and attempts to insert the peg at an incorrect position, leading to task failure despite a seemingly stable grasp. In the second case, the robot grasps the cylindrical peg with an imprecise hand posture, causing the thumb to slip during the transport phase. As a result, the peg deviates from the planned trajectory and misses the hole entirely.

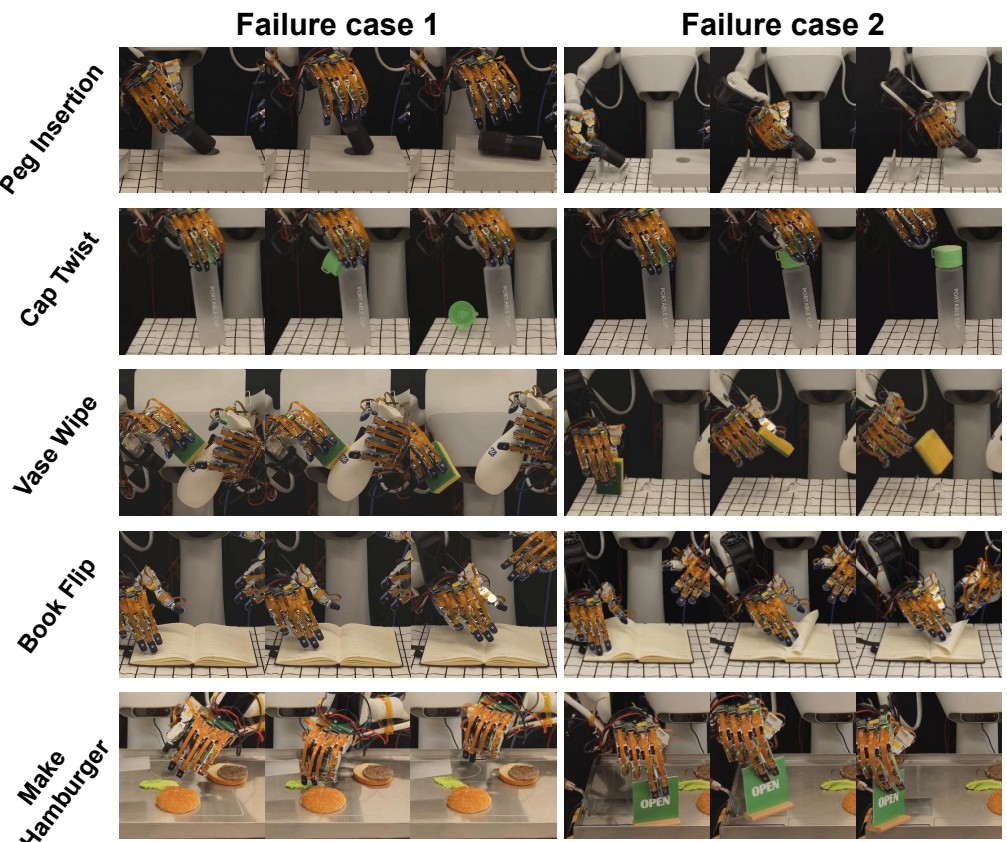

Figure 11: **Representative failure cases across all tasks.** Each row corresponds to one task, with two failure case sequences shown side by side.

### C.1.2  CAP TWIST

**Task Description**

The robot uses its right hand to rotate a cap off a bottle and place it on the table. The cap is initially tightened at a clockwise offset of about 100 degrees from the open position. Representative execution frames are shown in the second row of Fig. 10.

**Scoring Scheme**

Table 5: **Scoring criteria for Cap Twist.**

| Stage 1: Rotate (weight 2) | Description |
| --- | --- |
| 0 | No contact with the cap |
| 1 | Rotated 0–50° |
| 2 | Rotated 50–100°, or over-rotated |
| 3 | Fully unscrewed, cap held securely |
| **Stage 2: Place (weight 2)** | **Description** |
| 0 | Dropped immediately or stuck on bottle |
| 1 | Released before full separation |
| 2 | Partially placed or fell off |
| 3 | Stably placed on the table |

The task is divided into two stages: rotation and placement. Each is scored from 0 to 3, and a task is considered successful if the cap is fully unscrewed and placed stably (stage 1 score 3, stage 2 $\geq$2).

**Inference Results**

Table 6: **Cap Twist: inference results across models.**

| Model | Stage 1 | Stage 2 | HNS | Success Rate |
|---|---|---|---|---|
| DP | 1.1 | 0.3 | 0.23 | 0% |
| ACT | 2.4 | 1.1 | 0.58 | 40% |
| HATO | 1.8 | 0.5 | 0.38 | 10% |
| ACT w/T | 2.6 | 1.8 | 0.73 | 60% |
| Ours w/o CrossAttention | 2.9 | 2.2 | 0.85 | 70% |
| Ours w/o AutoRegressive | 3.0 | 2.3 | 0.88 | 70% |
| Ours w/o Two-Stage | 2.7 | 2.0 | 0.78 | 60% |
| Ours | 3.0 | 2.9 | 0.98 | 100% |

Table 6 presents the model performance on the cap twist task. Our method achieves the best HNS score (0.98) and 100% success rate, highlighting the advantage of fine-grained tactile reasoning.

**Failure Case Analysis**

In the second row of Fig. 11, two failure cases from the cap twist task are shown. In the first case, the robot fails to detect that the cap has already loosened and continues to apply torque unnecessarily, resulting in over-rotation that destabilizes the object. In the second case, the fingers lose contact during the twisting motion, leading to slippage and an insufficient rotation angle, which prevents the cap from being successfully removed.

C.1.3   VASE WIPE

**Task Description**

The robot uses its left hand to pick up a vase and its right hand to grasp a sponge. It then wipes away the blue ink mark located at the center of the vase. Representative execution frames are shown in the third row of Fig. 10.

**Scoring Scheme**

Table 7: **Scoring criteria for Vase Wipe.**

| Stage 1: Pick (weight 1) | Description |
|---|---|
| 0 | Failed to grasp the sponge |
| 1 | Grasped only a corner of sponge |
| 2 | Unstable grasp with partial control |
| 3 | Firm 3-finger grasp with full control |
| **Stage 2: Wipe (weight 2)** | **Description** |
| 0 | No contact with the ink mark |
| 1 | Wiped less than 50% |
| 2 | Wiped 50–90%, some ink remains |
| 3 | Fully wiped the ink area clean |

The task is divided into two stages: sponge grasping (pick) and vase wiping (wipe), both scored from 0 to 3. If the operator intervenes to re-adjust the vase grasp during stage 1, the score is reduced by 1. The task is considered successful only if both stages score 3.

**Inference Results**

Table 8 shows the quantitative performance on the vase wiping task. Our method again achieves the best HNS (0.98) and 90% success rate, showing reliable grasping and contact-driven wiping.

Table 8: **Vase Wipe: inference results across models.**

| Model | Stage 1 | Stage 2 | HNS | Success Rate |
|-------|---------|---------|-----|--------------|
| DP | 1.8 | 1.3 | 0.49 | 30% |
| ACT | 2.0 | 1.5 | 0.56 | 30% |
| HATO | 2.5 | 1.7 | 0.65 | 40% |
| ACT w/T | 3.0 | 1.9 | 0.75 | 40% |
| Ours w/o CrossAttention | 3.0 | 2.5 | 0.89 | 70% |
| Ours w/o AutoRegressive | 3.0 | 2.5 | 0.89 | 60% |
| Ours w/o Two-Stage | 2.1 | 1.6 | 0.59 | 40% |
| Ours | 3.0 | 2.9 | 0.98 | 90% |

**Failure Case Analysis**

The third row of Fig. 11 illustrates two typical failure modes in the vase wiping task. In the first case, the robot applies insufficient force during the wiping motion, resulting in incomplete surface contact between the sponge and the vase. Consequently, the ink mark is not fully removed. In the second case, excessive force is applied during the grasping phase, causing the sponge to slip out of the robot's fingers before the wiping action begins.

### C.1.4   BOOK FLIP

**Task Description**

The robot uses its right-hand middle finger to flip up a single page and then presses the page down using its left hand. Representative execution frames are shown in the fourth row of Fig. 10.

**Scoring Scheme**

Table 9: **Scoring criteria for Book Flip.**

| Stage 1: Flip (weight 2) | Description |
|--------------------------|-------------|
| 0 | No contact with the page |
| 1 | Touched but failed to lift / flipped multiple pages |
| 2 | Lifted halfway but stopped |
| 3 | Fully flipped one page |

| Stage 2: Press (weight 2) | Description |
|---------------------------|-------------|
| 0 | No contact with the page |
| 1 | Insufficient force, page rebounds |
| 2 | Pressed down, but misaligned |
| 3 | Fully and correctly pressed the page down |

This task includes two stages: flipping and pressing. Each stage is scored from 0 to 3. The task is considered successful if stage 1 scores 3 and stage 2 scores $\geq 2$.

**Inference Results**

Table 10 shows performance on the book flip task. Our method achieves the highest HNS (0.93) and 90% success rate, outperforming all baselines.

**Failure Case Analysis**

Figure 11, fourth row, presents two failure modes in the book flip task. In the first case, the robot fails to perceive the presence or precise location of the page edge, resulting in a poking motion that completely misses the page during the flipping attempt. In the second case, the robot applies excessive downward force before initiating the flip, which presses the page flat against the book and prevents it from being lifted.

Table 10: **Book Flip: inference results across models.**

| Model | Stage 1 | Stage 2 | HNS | Success Rate |
|---|---|---|---|---|
| DP | 1.5 | 0.5 | 0.35 | 10% |
| ACT | 1.9 | 0.7 | 0.43 | 20% |
| HATO | 2.0 | 0.6 | 0.43 | 30% |
| ACT w/T | 2.3 | 0.9 | 0.53 | 40% |
| Ours w/o CrossAttention | 2.7 | 2.1 | 0.80 | 70% |
| Ours w/o AutoRegressive | 2.7 | 1.9 | 0.77 | 70% |
| Ours w/o Two-Stage | 2.7 | 1.2 | 0.65 | 60% |
| Ours | 3.0 | 2.6 | 0.93 | 90% |

## C.2 LONG-HORIZON TASK: MAKE HAMBURGER

**Workspace Setup**

The long-horizon task is conducted on a customized metallic tabletop with seven designated ingredient/tool zones, as shown in Fig. 12. Each object is placed within either a circular or rectangular region marked on the tray. These regions serve as initialization zones with controlled spatial variability to support generalization. During both training and evaluation, each item is placed randomly within its assigned zone (up to 3cm positional jitter), ensuring that the policy must perform robust multimodal perception and execution.

**Task Description**

The long-horizon task involves a full hamburger assembly sequence requiring precise tool use and multi-stage coordination. The robot begins by flipping a wooden card from "closed" to "open" to indicate the start of service. It then uses its right hand to grasp a spatula and sequentially completes the following

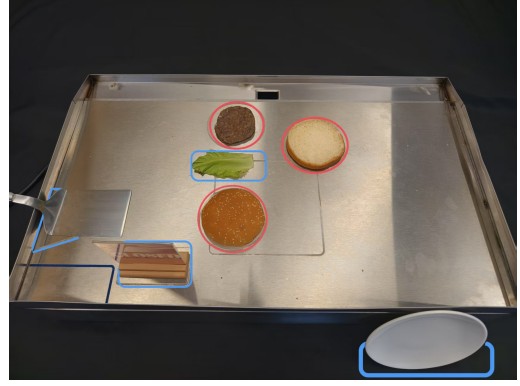

Figure 12: **Long-horizon task setup.** Seven components are placed in predefined zones—circular (ingredients) or rectangular (tools). Objects are randomly initialized within these areas to test spatial generalization.

steps: (1) lift and place the meat patty onto the bottom bread, (2) place a piece of lettuce, and (3) lift and place the top bread. Once the hamburger is assembled, the robot places it onto a plate handed over by a human. Finally, it returns the spatula to its original position and flips the sign back to "closed" to indicate task completion.

**Scoring Scheme**

The long-horizon hamburger task is decomposed into 11 sequential stages, covering symbolic interaction (sign flipping), tool use (spatula manipulation), ingredient assembly (meat patty, lettuce, bun), and final delivery. Each stage is scored from 0 to 3, where 0 indicates failure or no attempt, 1–2 denote partial or unstable execution, and 3 represents correct and stable completion. To better reflect task complexity and tactile sensitivity, each stage is assigned a specific weight: for example, sign flipping and deformable object handling (lettuce, bun) are given higher weights due to their reliance on fine-grained control and multi-finger dexterity.

The weighted stage scores are used to compute a Human Normalized Score (HNS), which reflects the overall task performance. A stage is considered successful if the score is at least 1. The entire task is marked as successful only when all 11 stages meet this threshold. Table 11 details the scoring criteria and weights for each stage.

**Failure Case Analysis**

Table 11: **Scoring criteria for the long-horizon hamburger task.**

| Stage | Action | Weight | Score Description |
|---|---|---|---|
| 1 | Flip sign (start) | 2 | 0: miss/fail; 1–2: partial (0–180°); 3: clean flip |
| 2 | Grab spatula | 2 | 0: miss; 1–2: unstable grasp; 3: secure grasp |
| 3 | Lift meat patty | 1 | 0: failed; 1–2: partial lift; 3: stable lift |
| 4 | Place meat patty | 1 | 0: miss; 1–2: partial/inaccurate; 3: centered |
| 5 | Grasp lettuce | 2 | 0: miss; 1–2: loose grasp; 3: stable placement |
| 6 | Lift top bread | 2 | 0: failed; 1–2: unstable or too forceful; 3: correct |
| 7 | Place top bread | 1 | 0: miss; 1–2: inaccurate; 3: clean stack |
| 8 | Lift hamburger | 1 | 0: failed; 1–2: unstable; 3: correct lift |
| 9 | Place on plate | 1 | 0: miss; 1–2: off-center; 3: perfect placement |
| 10 | Return spatula | 1 | 0: drop/fail; 1–2: misaligned; 3: accurate return |
| 11 | Flip sign (end) | 2 | 0: fail; 1–2: partial rotation; 3: clean close flip |

The fifth row of Fig. 11 shows two failure cases from the long-horizon hamburger assembly task. In the first case, the robot fails during stage 5 (grasping the lettuce): the grasp is unstable and incomplete, resulting in the lettuce slipping from the fingers before it can be placed. In the second case, the failure occurs in stage 1 (flipping the sign): although the sign is flipped, an incorrect grasp orientation causes the sign to rotate unintentionally during the movement, leading to a collision with the edge of the stove and blocking task progression.

## D  LIMITATION

Due to the inherent constraints of imitation learning, our policy lacks the capability to autonomously generalize to novel tasks. It still depends on human teleoperation for data collection. While our method demonstrates strong performance in long-horizon and challenging visuo-tactile manipulation tasks, its stability can be affected in scenarios where tactile feedback is less critical. This is primarily due to sensor noise and the limitations of the representation learning process. Future directions include improving generalization beyond demonstrations, reducing reliance on teleoperation, and enhancing robustness against sensor noise.

