# OpenReview forum: "ViTacFormer: Learning Cross-Modal Representation for Visuo-Tactile Dexterous Manipulation"
_ICLR.cc/2026/Conference — ICLR 2026 Conference Withdrawn Submission_

### Official Review · Reviewer_JtXN · 2025-10-30

**Soundness:** 3
**Presentation:** 3
**Contribution:** 2
**Rating:** 2
**Confidence:** 5

**Summary:**

Overall, while the engineering is appreciated, the experimental evaluation is not well supported or fully convincing. The learning contribution is limited. Prior work on teleoperation/data collection and behavior cloning across diverse representations already exists, which diminishes the novelty of this submission

**Strengths:**

Impressive real-world system integration; the long-horizon task showcases substantial engineering effort and careful execution.

**Weaknesses:**

Minor weaknesses

1. The paper appears better suited to robotics venues such as CoRL or RSS.

2. The integration of bimanual manipulation and tactile sensing is appreciated; however, the paper does not fully justify the necessity of tacitle in task selection or the necessity of bimanual hands.

Major weaknesses

1. The real-world engineering effort is substantial, but the contribution is not novel and does not constitute a comprehensive benchmark suite.

2. The vision–touch cross-attention mechanism is not novel (e.g., prior work on cross-modal representation); while its application to robotics is appreciated, the paper should clarify its contribution by comparing with existing cross-modal representation work.[1][2]

[1] LEARNING TO JOINTLY UNDERSTAND VISUAL AND TACTILE SIGNALS
[2] VITAS: VISUAL TACTILE SOFT FUSION CONTRASTIVE LEARNING FOR REINFORCEMENT LEARNING

3. The four short-horizon tasks in Fig. 4 do not adequately justify the need for tactile sensing or bimanual manipulation. For peg insertion, please report the clearance between the peg and socket. The vase-wipe and book-flip tasks do not clearly demonstrate the benefits of tactile sensing. Moreover, peg insertion, cap twist, and book flip are executed with a single hand and therefore do not motivate a bimanual setup.

4. Baseline comparisons and implementations are questionable. Please include stronger baselines using multimodal fusion policies (e.g., diffusion and ACT frameworks). In Table 1, results for the four short-horizon tasks are not compelling; running at least 20 trials would be more convincing. Also provide failure cases where the original Diffusion Policy achieves only 0–20% success.

5. Baseline comparison with other cross-modal representation work is missing. A comparison against recent cross-modal representation/fusion methods is missing. Please include baselines that explicitly model cross-modal alignment or attention to contextualize the claimed contribution.

6. The long-horizon demonstrations, while impressive, largely compose pick-and-place primitives and thus do not justify the use of tactile sensing. Please describe specific cases where the “w/o touch” baseline fails, especially since this task is highlighted as a key contribution in the introduction.

7. While the long horizon task is impressive, the paper does not fully justify why the proposed pipeline supports long-horizon success. An ablation separating data quality/quantity (e.g., teleop protocol, filtering, augmentation) from learning method (architecture, loss, training schedule) is needed to determine whether improvements stem from better data, better learning, or both.

**Questions:**

See weakness above.

---

### Official Review · Reviewer_wA2P · 2025-10-30

**Soundness:** 2
**Presentation:** 3
**Contribution:** 2
**Rating:** 4
**Confidence:** 5

**Summary:**

This paper set up a comprehensive robot hardware systems and introduces a novel visiual-tactile policy learning pipeline, which includes cross-attention, future tactile auto-regressive prediction, and two-stage training. To evaluate the effectiveness, this paper conduct 4 short-horizon tasks for comparing with other baseline and one 11-step long-horizon tasks for demonstrating its effectiveness.

**Strengths:**

1. This paper proposes a novel transformer-based visual-tactile policy learning pipeline, which includes next tactile prediction and two stage training.

2. This paper sets up four short-horizon dexterous and contact-rich manipulation tasks for evaluating the effectiveness of the proposed method, where it outperforms the baselines.

3. The performance for a 11-stage long-horizon dexterous manipulation task is impressive.

**Weaknesses:**

1. The major insight, a lack of an effective model that learns cross-modal representations for visuo-tactile dexterous manipulation, might be overclaimed. There are plenty of works work on cross-modal representation learning for visuo-tactile perceptions, including but not limited to [1, 2, 3], even together with language [4, 5]. Also, some of them already use cross-attention [1, 2] and autogressive curriculum learning [3] in multisensory policy learning. Discussing with those work and not declaring that this components are unique contributions might be helpful.

2. This pipeline can work for 11-stage long-horizon task is very impressive. However, it's not pretty clear why it works well. It has comprehensive ablations with short-horizon tasks but not this long-horizon task. Adding more ablations and show some analysis might be helpful.

3. Comparing with some missed SOTA visual-tactile diffusion-based policy learning pipeline might be important, such as reactive diffusion policy [6].

[1]. Li et al., See, Hear, Feel: Smart Sensory Fusion for Robotic Manipulation, CoRL 2022

[2]. Feng et al., Play to the Score: Stage-Guided Dynamic Multi-Sensory Fusion for Robotic Manipulation, CoRL 2024

[3]. Liu et al., FACTR: Force-Attending Curriculum Training for Contact-Rich Policy Learning, RSS 2025

[4]. Yang et al., Binding Touch to Everything: Learning Unified Multimodal Tactile Representations, CVPR 2024

[5]. Fu et al., A Touch, Vision, and Language Dataset for Multimodal Alignment, ICML 2024

[6]. Han et al., Reactive Diffusion Policy: Slow-Fast Visual-Tactile Policy Learning for Contact-Rich Manipulation, RSS 2025

**Questions:**

1. The future tactile prediction is very interesting. However, there are some other work using world model to predict future observations [1], or doing next token prediction [2], object motion prediction [3, 4], or using future actions [5] as conditions of diffusion policies. Why only using future tactile prediction but not other observations or actions?

2. The raw tactile signal are 320*240 tactile images. How to change them into 3 axes contact force? Doesn't different contact position on the tactile sensor matter for policy learning?

[1]. Hafner et al., Mastering Diverse Domains through World Models, Nature

[2]. Fu et al., In-Context Imitation Learning via Next-Token Prediction, ICRA 2025

[3]. Yu et al., GenFlowRL: Shaping Rewards with Generative Object-Centric Flow in Visual Reinforcement Learning, ICCV 2025

[4]. Su et al., Motion Before Action: Diffusing Object Motion as Manipulation Condition, RA-L

[5]. Ranawaka et al., SAIL: Faster-than-Demonstration Execution of Imitation Learning Policies, CoRL 2025

---

### Official Review · Reviewer_xePd · 2025-10-31

**Soundness:** 2
**Presentation:** 2
**Contribution:** 2
**Rating:** 2
**Confidence:** 4

**Summary:**

The paper presents ViTacFormer, a transformer-based framework for visuo-tactile policy learning in dexterous manipulation. It fuses visual and tactile modalities through cross-attention and introduces an autoregressive tactile forecasting module that predicts future tactile feedback to inform action generation. Built on a conditional variational autoencoder, the model captures high-level action intent and is trained with a two-phase curriculum that transitions from ground-truth to predicted tactile inputs. Experiments on several real-robot tasks, including long-horizon manipulation, show improvements over imitation-learning baselines such as ACT, HATO, and Diffusion Policy. While the system demonstrates strong empirical performance and real-world robustness, its conceptual contribution lies mainly in architectural integration rather than methodological innovation.

**Strengths:**

**Strengths:**

**1. Good Writing:** The paper is clear, concise, and well-organized. Its explanations of tasks and evaluation approach are logically structured, facilitating straightforward comprehension of the methodology and results.

**2. Experiments:** The experimental evaluation is a major strength. It is comprehensive and well-executed, featuring real-robot experiments on multiple contact-rich manipulation and  long-horizon tasks with empirical improvements, which is impressive.

**3. Framework:** The work contributes a practically functional and stable visuo-tactile policy framework, showing that integrating predictive tactile signals into imitation learning can be achieved reliably on physical systems

**Weaknesses:**

**Weaknesses:**

**1. Novelty:** Conceptually, ViTacFormer is a  composition of known ideas like cross-attention for multimodal fusion, a CVAE/ACT-style latent policy, autoregressive tactile forecasting, and a simple two-phase curriculum rather than a new learning principle. The proposed two-phase curriculum training with ground-truth tactile inputs appears largely heuristic and a lacks justification. The paper provides almost no in-depth analysis of why this ratio works, how it compares to alternatives, or how it relates to existing curriculum learning literature. Given this, the conceptual advance is incremental: the work operationalizes established components in a pipeline, but does not introduce a new objective, estimator, or theoretical mechanism for cross-modal representation learning.

**2. Representation Comparison:** Although ViTacFormer positions itself as a cross-modal representation-learning framework that unifies vision and touch for dexterous control, the paper provides no direct evaluation or comparison of its learned representations. The experiments focus solely on end-to-end task success, without any analysis of representation quality (on standard benchmarks) such as linear-probe tests, frozen-encoder transfer, or cross-modal retrieval, which are commonly used in visual-tactile papers [1-5]. Although removing cross-attention reduces performance as shown in ablation study, that only reveals that the fusion module matters, not that the learned representations are strong or task-general. Evaluating representation quality (e.g., linear probes) or comparing to relevant baselines would substantiate the representation-learning claim. Without such controlled comparisons, the paper’s representation claim remains qualitative rather than analytical.

**3. Ablation Study:** While the paper presents an ablation study removing individual components like cross-attention, tactile forecasting, the CVAE latent, and the two-phase curriculum, the analysis is not designed to reveal causal contributions. Showing that performance drops when a component is removed demonstrates correlation but does not establish causality: we learn that the module helps, but not why or how much (the in-depth analysis seems to be missing). For example, removing the cross-attention block would reduces model capacity. How can we know that the performance drop is not due to that? Without introducing an equal-parameter control (maybe MLP of the same size), it’s unclear whether the improvement stems from better modality fusion or simply from having more parameters. Likewise,  the curriculum is a fixed ratio, is not varied, so we cannot tell whether this exact split is important or if any gradual schedule would suffice. Collectively, the study confirms that the full model works best but the analysis of their contribution needs to be better.

**References**

[1] Yang, Fengyu, et al. "Binding touch to everything: Learning unified multimodal tactile representations." Proceedings of the IEEE/CVF Conference on Computer Vision and Pattern Recognition. 2024.

[2] Dave, Vedant, Fotios Lygerakis, and Elmar Rueckert. "Multimodal visual-tactile representation learning through self-supervised contrastive pre-training." 2024 IEEE International Conference on Robotics and Automation (ICRA). IEEE, 2024.

[3] Huang, Binghao, et al. "3d-vitac: Learning fine-grained manipulation with visuo-tactile sensing." arXiv preprint arXiv:2410.24091 (2024).

[4] Fu, Letian, et al. "A touch, vision, and language dataset for multimodal alignment." arXiv preprint arXiv:2402.13232 (2024).

[5] Cheng, Ning, et al. "Touch100k: A large-scale touch-language-vision dataset for touch-centric multimodal representation." Information Fusion (2025): 103305.

**Questions:**

They are mentioned in the Weaknesses.

---

### Official Review · Reviewer_a92M · 2025-11-01

**Soundness:** 3
**Presentation:** 3
**Contribution:** 3
**Rating:** 4
**Confidence:** 4

**Summary:**

This paper, "ViTacFormer," presents a novel approach for learning cross-modal visuo-tactile representations to enhance real-world dexterous manipulation using anthropomorphic hands. The core contribution is a unified framework built upon a conditional VAE (based on ACT [1]) that deeply fuses high-resolution visual and tactile observations. An Autoregressive tactile-prediction head and a two-phase curriculum strategy were adopted, enabling the model to effectively model the dexterous touch dynamics. A complex dual-arm, high-DoF robot system is set up and utilized for data collection and real-robot experiments. The authors validated ViTacFormer's significant performance advantage over baselines on 4 short-horizon tasks and 1 long-horizon task. In particular, the first super long-horizon dexterous manipulation demo (2.5 min with 11 sequential stages to make a hamburger) is highly impressive.

**Strengths:**

1. **High-Fidelity Hardware and Data Setup:** The construction of a sophisticated real robotic system featuring active vision, dense tactile sensing, and high-DoF capabilities is commendable. This system was effectively used for both data collection (via human teleoperation) and rigorous experimental validation.

2. **State-of-the-Art Real-World Performance:** Achieving a nearly 50% improvement in success rate over competitive state-of-the-art baselines like HATO and ACTw/T on fine-grained, contact-rich tasks (Cap Twist, Book Flip) is a compelling achievement that validates the visuo-tactile fusion method.

3. **Impressive Demonstration on Long-Horizon Tasks:** Successfully completing the 11-stage, 2.5-minute "Make Hamburger" task on a real, high-DoF anthropomorphic system represents a major milestone in real-world dexterous manipulation and is an excellent showcase of the framework's robustness and precision.

**Weaknesses:**

1. **Limited Algorithmic Novelty:**
   - Cross attention for multimodal fusion is a very common practice[2, 3, 4, 5].
   - Autoregressively predicting the next timestep is also a basic practice particularly in the natural language process[6] and computer vision[7].

2. **Insufficient Validation of Tactile Signal Forecasting:** The paper lacks qualitative and quantitative experiments regarding the Tactile Signal Forecasting component. For example, supplementary experiments illustrating the prediction accuracy of the next-timestep tactile signal are necessary to justify its effectiveness.

3. **Limited Data Scale and Generalizability:** The policy is trained on a small dataset of only 50 expert trajectories per task. While the system generalizes to slight positional perturbations, the reliance on such a limited dataset raises concerns about the generalizability of the learned representation to wider variations in objects, environments, or task conditions outside the demonstrated scenarios.

[1] Learning Fine-Grained Bimanual Manipulation with Low-Cost Hardware
[2] Unified Vision-Language Pre-Training for Image Captioning and VQA
[3] ViLBERT: Pre-training Task-Agnostic Visiolinguistic Representations for Vision-and-Language Tasks
[4] LXMERT: Learning Cross-Modality Encoder Representations from Transformers
[5] DALL.E: Zero-Shot Text-to-Image Generation
[6] Improving Language Understanding by Generative Pre-Training
[7] Conditional Image Generation with PixelCNN Decoders

**Questions:**

see weakness

---

### Note · Authors · 2025-11-12

I have read and agree with the venue's withdrawal policy on behalf of myself and my co-authors.